# High Rate Performance Supercapacitors Based on N, O Co-Doped Hierarchical Porous Carbon Foams Synthesized via Chemical Blowing and Dual Templates

**DOI:** 10.3390/molecules28196994

**Published:** 2023-10-09

**Authors:** Qian Zhang, Li Feng, Zhenlu Liu, Longjun Jiang, Tiancheng Lan, Chunmei Zhang, Kunming Liu, Shuijian He

**Affiliations:** 1Co-Innovation Center of Efficient Processing and Utilization of Forest Resources, International Innovation Center for Forest Chemicals and Materials, College of Science, Nanjing Forestry University, Nanjing 210037, China; hzndl17@163.com (L.J.); 18779010213@163.com (T.L.); 2College of Materials Science and Engineering, Nanjing Forestry University, Nanjing 210037, China; fengli_better@163.com (L.F.); liu17864279694@163.com (Z.L.); 3Institute of Materials Science and Devices, School of Materials Science and Engineering, Suzhou University of Science and Technology, Suzhou 215009, China; cmzhang@usts.edu.cn; 4Faculty of Materials Metallurgy and Chemistry, Jiangxi University of Science and Technology, Ganzhou 341000, China; liukunming@jxust.edu.cn

**Keywords:** biomass products, supercapacitor, N, O Co-Doping, dual template, rate performance

## Abstract

N, O Co-Doped porous carbon materials are promising electrode materials for supercapacitors. However, it is still a challenge to prepare high capacitance performance N, O Co-Doped porous carbon materials with balanced pore structure. In this work, a simple chemical blowing method was developed to produce hierarchal porous carbon materials with Zn(NO_3_)_2_·6H_2_O and Fe(NO_3_)_3_·9H_2_O as the foaming agents and precursors of dual templates. Soybean protein isolate served as a self-doping carbon source. The amount of Fe(NO_3_)_3_·9H_2_O influenced the microstructure, element content and capacitance performance of the obtained porous carbon materials. The optimized sample CZnFe-5 with the addition of 5% Fe(NO_3_)_3_·9H_2_O displayed the best capacitance performance. The specific capacitance reached 271 F g^−1^ at 0.2 A g^−1^ and retained 133 F g^−1^ at 100 A g^−1^. The CZnFe-5//CZnFe-5 symmetric supercapacitors delivered a maximum energy density of 16.83 Wh kg^−1^ and good stability with capacitance retention of 86.33% after 40,000 cycles tests at 50 A g^−1^. The symmetric supercapacitors exhibited potential applications in lighting LED bulbs with a voltage of 3 V. This work provides a new strategy for the synthesis of hierarchical porous carbon materials for supercapacitors from low-cost biomass products.

## 1. Introduction

With the worsening energy shortage and environmental crisis, the search for sustainable energy sources and their efficient energy-storage devices has become an urgent problem for researchers [1,2,3]. Recently, various advanced energy-storage devices have been developed [4,5,6,7,8]. Supercapacitors are promising energy-storage devices which fill in the gap of high-power-density traditional capacitors and high-energy-density secondary batteries [9]. Supercapacitors have been applied in the power grid, transportation, regenerative braking, wearable electronic products, etc. Hierarchical porous carbon materials are widely applied as electrode materials for supercapacitors [10,11,12]. Micropores provide high specific surface area (SSA) for the formation of an electrical double layer, which is the basis of high capacitance. Mesopores and macropores serve as mass-transfer channels which are important for the rate performance. Engineering the pore structure can boost the capacitance performance of porous carbon materials [13].

Various precursors have been utilized to synthesis porous carbon materials [14,15,16,17,18]. Among those precursors, biomass attracts great attention due to advantages such as abundance, low cost and renewability [19,20,21]. Moreover, biomass contains heteroatoms like N, O, P and S which are beneficial for the synthesis of heteroatom-doped carbon materials [22,23,24]. However, the specific surface area of carbonaceous products obtained from direct carbonization of biomass is low and the pore structure needs to be adjusted [25,26].

Our group has chosen soybean protein isolate (SPI) as the precursor to synthetic carbon materials for supercapacitors due to its high content of N and O [27,28]. Dense carbon particles with irregular morphology were obtained by direct carbonization of SPI, which showed a low SSA of 11.61 m^2^ g^−1^ and poor capacitance performance (46.3 F g^−1^ at 0.1 A g^−1^) [27]. Large-size carbon sheets were obtained when SPI and ZnCl_2_ were pyrolyzed together. When the mass ratio of SPI to ZnCl_2_ was 1:2, the obtained NC-3 sample had a maximum SSA of 1034.33 m^2^ g^−1^ and a relative high specific capacitance of 191.15 F g^−1^ at 0.1 A g^−1^. However, the thick sheet structure was detrimental to rate performance of NC-3 as the specific capacitance only retained 47.1% at 35 A g^−1^. To enhance the rate performance of SPI-derived carbon materials, ZnCl_2_ was replaced by Zn(NO_3_)_2_·6H_2_O and mesopore-dominated hierarchical porous carbon foams were obtained [28]. When the mass ratio of SPI to ZnCl_2_ was 2:3, the sample NC-1.5 obtained from a carbonization temperature of 850 °C had a maximum SSA of 1350.34 m^2^ g^−1^ and a higher specific capacitance of 248 F g^−1^ at 0.5 A g^−1^. The rate performance of NC-1.5 was enhanced as the specific capacitance retained 109 F g^−1^ at 100 A g^−1^.

In this work, Zn(NO_3_)_2_·6H_2_O is partly replaced by Fe(NO_3_)_3_·9H_2_O to generate highly interconnected porous carbon foam with the goal of further enhancing the rate performance of SPI-derived porous carbon materials. The intermediate products originated from decomposition of Fe(NO_3_)_3_·9H_2_O would etch the walls of carbon foam, facilitating ion transfer in the neighbor cells. This is because the obtained porous carbon foams exhibit enhanced capacitance performance compared to that of the counterpart synthesized from Zn(NO_3_)_2_·6H_2_O and SPI. The optimized sample shows 271 F g^−1^ at 0.2 A g^−1^ and retains 133 F g^−1^ at 100 A g^−1^. This research provides a new strategy to enhance the rate performance of hierarchical porous carbon materials for supercapacitors.

## 2. Results and Discussion

The preparation process for porous carbon foam is depicted in Figure 1. The morphology and structure of the prepared porous carbon foams can be revealed by FESEM and HRTEM. From Figure 2a–d, it can be seen that the SPI-derived carbonaceous materials end up in interconnected porous carbon sheets with the assistance of the chemical foaming effect. The gas mixture generated by the decomposition of Zn(NO_3_)_2_·6H_2_O and Fe(NO_3_)_3_·9H_2_O blisters the molten SPI to form an interconnected foam structure with abundant macropores [28]. ZnO and Fe_2_O_3_ nanoparticles were produced from the decomposition of Zn(NO_3_)_2_·6H_2_O and Fe(NO_3_)_3_·9H_2_O, which served as hard templates to generate mesopores and macropores. ZnO nanoparticles were reduced to Zn nanoparticles at elevated temperature by the carbothermal reduction [29]. Zn nanoparticles partly evaporated and generated rich micropores in the carbon foam when the temperature reached above 700 °C. Fe_2_O_3_ nanoparticles could be reduced to iron carbide and Fe nanoparticles by the carbothermal reduction at elevated temperature [30]. Iron carbide and Fe nanoparticles would further etch the carbon matrix, generated more nanopores on the carbon sheets and penetrated adjacent cells, as evident from Figure 2c. The foam structure would collapse if too much Fe(NO_3_)_3_·9H_2_O was added, as shown in Figure 2d. HRTEM images (Figure 2e,f) reveal the partly graphitized structure of carbon materials, which is attributed to the catalytic effect of Zn and Fe nanoparticles.

XRD patterns and Raman spectra were collected to characterize the crystal structure and the graphitization degree of the carbon materials. As shown in Figure 3a, XRD diffraction peaks at 25° and 43° correspond to (002) and (101) crystallographic planes of carbon materials, respectively [31]. The G band (~1587 cm^−1^) and D band (~1315 cm^−1^) in Raman spectra (Figure 3b) correspond to the in-plane vibration of ordered hybridized carbon architecture (sp^2^) and defect disordered frameworks (sp^3^), respectively. The intensity ratio of the two bands (I_D_/I_G_) reflects the graphitization degree of carbon material [32,33]. The I_D_/I_G_ for CZn, CZnFe-2.5, CZnFe-5 and CZnFe-7.5 is 1.51, 1.54, 1.52 and 1.57, respectively, indicating that the addition of a small amount of iron nitrate does not enhance the graphitization degree of as-obtained porous carbon materials.

The pore characteristics of carbon material play an important role in its capacitive performance. N_2_ adsorption–desorption isotherms can reveal the pore structure of the samples. As shown in Figure 3c,d, the samples all exhibit the combined feature of typical type I and type IV isotherm characteristics [34,35]. At the lower relative pressures (P/P_0_ < 0.1), the adsorption isotherms show a rapidly increasing trend, reflecting the presence of micropores in the carbon material [36]. When the relative pressure increases from 0.45 to 0.9, the isotherms all exhibit a clear H4 hysteresis line, which reveals the presence of a great amount of large mesopores [35,37]. The sudden increase in adsorption capacity at high relative pressure above 0.9 indicates that macropores also occupy a certain proportion of the carbon material [38]. The pore size distribution curves are shown in Figure 3d, which clearly display the hierarchical porous feature of as-obtained carbon materials. The porosity was greatly influenced by the addition of Fe(NO_3_)_3_·9H_2_O. In particular, the content of nanopores with the size larger than 1.0 nm was reduced, which leads to a sharp decrease in SSA. It can be seen from Table 1 that when Fe(NO_3_)_3_·9H_2_O was added, the specific surface area showed a significant decrease. However, the comparison revealed that the introduction of a trace amount of Fe(NO_3_)_3_·9H_2_O could increase the SSA and volume proportion of micropores in the pore structure. With the increase in Fe(NO_3_)_3_·9H_2_O, the specific surface area of CZnFe-X decreases first and then increases, and the average pore diameter (Da) increases from 2.32 nm to 2.69 nm. After the addition of Fe^3+^ salt and the dispersion of Zn particles after high-temperature pyrolysis and reduction, the double template formed by the two is removed by acid, and the etching effect is stronger than that of pure Zn(NO_3_)_2_·6H_2_O. The ratio of micropore volume (V_micro_) to total pore volume (V_t_) decreased from 49% for CZnFe-2.5 to 39% for CZnFe-5 and 37% for CZnFe-7.5, while the corresponding V_meso_ ratio increased from 42% to 50%. This is because the micropores in the carbon matrix gradually expand to mesopores [39,40].

XPS analysis was carried out to reveal the influence of Fe(NO_3_)_3_·9H_2_O on the heteroatom content and functional groups of resulting porous carbon material. The three distinct peaks in Figure 4a correspond to the three elements of C, N and O. Figure 4b shows the content of C, N and O in different samples. The N and O contents of the samples were slightly increased by adding 2.5% Fe(NO_3_)_3_·9H_2_O but decreased when 5% Fe(NO_3_)_3_·9H_2_O was added. Figure 4c shows the high-resolution spectra of N 1 s. The four fitted peaks represent pyridine nitrogen (N-6), pyrrole nitrogen (N-5), graphitic nitrogen (N-Q) and nitrogen oxide (N-O). Among them, N-6 and N-5 show strong electrochemical activity in aqueous electrolytes and are the main sources of pseudocapacitance for N-doped hierarchical porous carbon materials [41]. N-Q not only promotes electron transfer but also changes the overall electron distribution and increases the surface area of the active site through the doping atoms in the carbon–nitrogen network [42]. N-O enhances the hydrophilicity of the electrode material [43]. As shown in Figure 4d, the proportion of N-Q gradually increases with the increase in Fe content, which indicates that the incorporation of Fe(NO_3_)_3_·9H_2_O is conducive to the formation of N-Q, promotes electron transfer and improves capacitance performance [44]. The high-resolution spectra of O 1 s (Figure 4e) reflect the evolution trend of oxygen-containing functional groups. Three types of oxygen elements, C=O, C-O-C/C-OH and O-C=O, are specifically fitted [45]. The introduction of Fe^3+^ greatly increased the proportion of O-C=O. Sufficient oxygen-containing functional groups can improve the wettability of the carbon material by the electrolyte and reduce the internal resistance by changing the polarity of the surface [46]. At the same time, Faraday pseudocapacitance is introduced by oxygen-containing functional groups to improve the capacitive performance of carbon material.

The capacitance performance of the electrode materials was first evaluated by assembling a three-electrode system in 6 M KOH electrolyte. As shown in Figure 5a, the cyclic voltammetry (CV) curves of CZnFe-X are quasi-rectangular at a scan rate of 100 mV s^−1^, indicating that the electric double-layer capacitance plays a dominant role. The distortion of the curves is caused by the redox reaction of oxygen/nitrogen doping. The galvanostatic charge–discharge (GCD) curves obtained for all the electrodes tested at low current densities (0.5 A g^−1^) exhibit typical triangles and linear discharge slopes [47], further indicating their desirable capacitive properties (Figure 5b). At low current densities of 0.2 A g^−1^, the specific capacitances of CZn, CZnFe-2.5, CZnFe-5 and CZnFe-7.5 were 225 F g^−1^, 248 F g^−1^, 271 F g^−1^ and 233 F g^−1^, respectively. The capacitance is severely degraded with the increase in current density due to the insufficient contribution of pseudocapacitance from heteroatom-containing functional groups at limited time [48]. The capacitance remains stable above 5 A g^−1^ as electric double-layer capacitance could be formed in a very short time. CZnFe-5 exhibits the best rate performance among all the samples, retaining 133 F g^−1^ at 100 A g^−1^ (Figure 5c). The good rate performance of CZnFe-5 is attributed to the fast ion transport in the interconnected porous carbon matrix [49]. The capacitance retention of the CZnFe-5 electrode reaches 93.40% after 40,000 cycles at a high current density of 100 A g^−1^, indicating excellent electrochemical stability (Figure 5d). The good capacitive performance of CZnFe-5 is attributed to the synergistic effect of the hierarchical porous structure and heteroatom doping. Among them, the interconnected nanostructure shortens the ion diffusion distance, the mesopores provide ion transport pathways and the micropores can serve as ion-storage sites with high charge storage capacity [50]. Together with the redox effect caused by heteroatomic functional groups, CZnFe-5 exhibits good capacitance performance.

Electrochemical impedance spectra (EIS) were collected to reveal the resistance of porous carbon materials [51]. The diameter of the semicircle in the high-frequency region (Figure 5e) represents the charge transfer resistance (R_ct_), and the intercept with the real axis represents the series resistance (R_s_) [52]. The low-frequency region shows a vertical line related to the pure capacitive behavior [53]. The R_s_ is less than 1.2 Ω, indicating a low resistance between the interface of electrode/electrolyte. The heteroatomic doping of the material contributes to good surface wettability and low R_s_ [54]. The CZnFe-5 electrode has the smallest charge transfer resistance with the fastest ion transport and the lowest internal resistance. In the Bode plot (Figure 5f), all the samples have a phase angle close to −90° at low frequency, which means that they have typical capacitive properties [55]. The relaxation time constant τ_0_ (τ_0_ = 1/*f*_0_, *f*_0_ = 45 Hz) can be used to evaluate the discharge rate of the electrodes, and the value of τ_0_ for CZnFe-5 is 1.59 s, which is smaller than that of the other electrodes. Therefore, CZnFe-5 shows the best rate performance at larger charge and discharge current densities.

Electrochemical reaction kinetics were analyzed to reveal the charge storage mechanism of hierarchical porous carbon materials with CZnFe-5 as an example. Typically, the capacitive energy storage mechanisms for porous carbon materials are divided into diffusion-controlled capacitance (pseudocapacitance) and surface-controlled capacitance (electric double layer capacitance). The capacitive contribution of diffusion-controlled capacitance and surface-controlled capacitance can be calculated by the following empirical equations [56]:(1)$$i=av^{b}$$
(2)logi=blogv+loga
where i and *v* denote the peak current and scan rate, respectively, and a and b are variables. a and b values can be obtained from the function between *logi* and *logv*. When *b* = 1, the surface-controlled capacitive behavior dominates the capacitance. When *b* = 0.5, it is the diffusion-controlled process that dominates. When the value of b is between 0.5 and 1, it can be considered as a hybrid mechanism. As shown in Figure 6a, the b values obtained by fitting the current at −0.9 V from CV curves obtained at different scan rates are 0.82 and 0.92, demonstrating that the energy-storage mechanism of CZnFe-5 is a combination of electric double-layer capacitance and pseudocapacitance.

The capacitance contribution of the CZnFe-5 electrode is calculated from the Dunn’s equation [57]:(3)$$i(v)=k_{1}v+{k_{2}v}^{0.5}$$
where i (*v*) is the response current at a certain voltage; *k*_1_ and *k*_2_*v* are variables, and *k_1_v* and *k*_2_*v*^0.5^ represent the fast kinetic process for surface control and the slow kinetic process for diffusion control, respectively [58]. The capacitance contributions of the surface control process and the diffusion control process fitted to the CZnFe-5 electrode at different scan rates are depicted in Figure 6b. The contribution of surface control capacitance increases from 73.73% to 88.25% with the scan rate rise from 5 mV s^−1^ to 100 mV s^−1^. This result indicates that at low current densities, the heteroatomic functional groups contribute part of the pseudocapacitance through Faraday redox reactions and overall surface capacitive behavior dominates the charge storage [59]. The significant decrease in the contribution of pseudocapacitance at high scan rates is due to insufficient diffusion time. On the other hand, the high contribution of surface-controlled capacitance indicates that the CZnFe-5 electrode has good rate performance and surface charge storage capacity [60].

To further evaluate the practical application of the CZnFe-5 sample, the CZnFe-5//CZnFe-5 symmetric supercapacitors were assembled and tested with 6 M KOH as the electrolyte. The quasi-rectangular shape was well maintained when the scan rate was increased to 100 mV s^−1^, indicating outstanding EDLC characteristics and good electrochemical reversibility. As shown in Figure 7b, the GCD curves at different current densities showed symmetric triangles, further confirming its typical EDLC behavior. The GCD curves at different current densities were calculated to obtain the rate performance plot of CZnFe-5 (Figure 7c). Its specific capacitance is 122 F g^−1^ at a current density of 0.2 A g^−1^, and the capacitance retains 50% at 50 A g^−1^, indicating its good rate performance. The rate performance of CZnFe-5 is better than that of the commercial sample YP50F, with close mass loading of 2.7 mg (Appendix A). Figure 7d depicts the Nyquist plot of CZnFe-5//CZnFe-5, which shows that it has a low R_s_ (~2 Ω) and a small charge transfer resistance R_ct_ (1.3 Ω), implying its excellent ionic conductivity and high charge transfer rate in the electrolyte. Moreover, it was able to achieve 88.14% capacitance retention after 10,000 cycles at 2 A g^−1^ (Appendix A) and 86.33% after 40,000 cycles at 50 A g^−1^ (Figure 7e), demonstrating good stability. To demonstrate the practical application of symmetrical devices, 22 green LEDs of 3 V are easily lit by three CZnFe-5//CZnFe-5 symmetrical devices powered in series. The Ragone plot in Figure 7f shows that the maximum energy density of the symmetric SCs is 16.83 Wh kg^−1^ with a power density of 199.36 W kg^−1^. Even at the maximum power density of 8.56 kW kg^−1^, the energy density remains at 9.73 Wh kg^−1^, which is comparable to other reported carbon materials in the last three years [61,62,63,64,65,66,67].

## 3. Materials and Methods

### 3.1. Chemicals

All reagents were used directly without further purification. HNO_3_, KOH, Zn(NO_3_)_2_·6H_2_O and Fe(NO_3_)_3_·9H_2_O were ordered from Nanjing Chemical Reagent Co., Ltd. (Nanjing, China). SPI was obtained from Macklin Co., Ltd. (Nanjing, China).

### 3.2. Synthesis of CZnFe-X Hierarchical Porous Carbon

The preparation process for porous carbon foam is depicted in Figure 1. The mass ratio of SPI to nitrate was fixed at 2:3, and the weight percentage of Fe(NO_3_)_3_·9H_2_O to Zn(NO_3_)_2_·6H_2_O was varied from 2.5% to 5% and 7.5%. Firstly, 3 g of the mixed nitrates was dissolved in 10 mL deionized water with stirring. After complete dissolution, 2 g of SPI was added and sonicated at 60 Hz for 30 min. As-prepared yellow slurry was transferred into a porcelain boat and dried in a blast oven at 100 °C. The dried samples were directly transferred to a tube furnace and carbonized in N_2_ at 850 °C for 2 h with a heating rate of 5 °C min^−1^. To remove the Fe nanoparticles from the samples, 0.2 g of the carbonized samples was mixed with 30 mL of 0.1 M HNO_3_ solution and sealed in a Teflon-lined autoclave, heating at 120 °C for 6 h. After the autoclave was completely cooled to room temperature, the black solids were extracted, rinsed repeatedly with deionized water until the pH of the filtrate reached 7. The dried sample was defined as CZnFe-X (where X is the percentage of Fe(NO_3_)_3_·9H_2_O). The control sample was prepared in the same procedure with 3 g Zn(NO_3_)_2_·6H_2_O and labeled as CZn.

### 3.3. Characterization Technique

The crystal structure of the samples was explored on an X-ray diffractor (XRD, Ultima IV, Rigaku, Japan) with point detector and Cu Kα radiation at a scan speed of 10° min^−1^ and step width of 0.02° [68,69]. Raman spectra were collected on a Raman laser spectrum analyzer (DXR532, Thermo Fisher, Carlsbad, CA, USA). The micromorphology of samples was revealed by field emission scanning electron microscopy (FESEM, JSM-7600F, Japan Electronics Co., Ltd., Tokyo, Japan) and transmission electron microscopy (TEM, JEM-2100, Japan Electronics Co., Ltd., Tokyo, Japan). The surface chemistry analysis was conducted by using X-ray photoelectron spectroscopy (XPS, Thermo Scientific, St. Louis, MI, USA). XPS test was performed on one spot for each sample. One batch of samples was tested. Pore structure analysis was carried out on a Micromeritics ASAP 2460. Prior to the test, the samples were degassed under vacuum at 200 °C for 8 h. The total SSA was obtained by the Brunauer–Emmett–Teller (BET) method. The micropore specific surface area (S_micro_), mesopore specific surface area (S_meso_), micropore volume (V_micro_), mesopore volume (V_meso_) and total pore volume (V_t_) were obtained from the t-plot method.

### 3.4. Electrochemical Testing

Porous carbon foam, conductive carbon black and polyvinylidene fluoride were mixed at the mass ratio of 8:1:1 with an appropriate amount of N-methylpyrrolidone and ground into a uniform slurry. The slurry was coated on nickel foams with a coating area of about 1 cm × 1 cm, and then, the nickel foam was dried in a vacuum oven at 60 °C for 12 h. Finally, the electrodes were compacted by a tablet press at a pressure of 10 MPa. The mass of active material loaded onto each piece of electrode was about 2.5 mg. Electrodes made from the commercial sample YP50F with close mass loading of 2.7 mg following the same procedure were used for comparison.

The electrochemical performance of the assembled three-electrode system and symmetric supercapacitor was measured by an electrochemical workstation (CHI 760E, Chenhua, Shanghai, China) in 6 M KOH electrolyte at room temperature. The CZnFe-X electrodes were used as the working electrode and the counter electrode along with a saturated calomel electrode (Hg/Hg_2_Cl_2_, SCE) as the reference electrode to form the three-electrode system. A piece of cellulose paper (NKK-MPF30AC-100) was taken as the separator. The symmetrical supercapacitors were assembled by two CZnFe-5 electrodes.

The specific capacitance (C, F g^−1^) for the three-electrode system is calculated by Equation (4). The specific capacitance (Cs, F g^−1^) of a single electrode in the two-electrode symmetrical system is calculated by Equation (5). The energy density and power density of the symmetrical supercapacitor are calculated by Equations (6) and (7), respectively.
(4)$$C=\frac{I\times \Delta t}{m\times \Delta V}$$
(5)$$C_{s}=2\;\times \;\frac{I\times \Delta t}{m\Delta V}$$
(6)$$E=\frac{1}{2}\times C_{s}\Delta V^{2}\times \frac{1}{3.6}$$
(7)$$P=\frac{3600\times E}{\Delta t}$$
where I (A), Δt (s), m (g) and ΔV (V) represent the discharge current, discharge time, mass of active material loaded on a single electrode and potential window (excluding IR drop), respectively. E (Wh kg^−1^) refers to the energy density, and P (W kg^−1^) represents the power density.

## 4. Conclusions

In summary, N, O Co-Doped hierarchical porous carbon foams with enriched micropores were prepared by using soybean protein isolate as the carbon source and two nitrates, Fe(NO_3_)_3_·9H_2_O and Zn(NO_3_)_2_·6H_2_O, as the foaming agents and hard templates. The microstructure, element content and capacitance performance of the resulting products were influenced by the content of Fe(NO_3_)_3_·9H_2_O. The iron carbide and iron nanoparticles obtained from decomposition of Fe(NO_3_)_3_·9H_2_O had an etch effect on the carbon framework, which facilitated the electrolyte transfer in carbon foam. However, an excess amount of Fe(NO_3_)_3_·9H_2_O caused the collapse of the 3D interconnected foam structure. The porosity was greatly influenced by the addition of Fe(NO_3_)_3_·9H_2_O. In particular, the content of nanopores with the size larger than 1.0 nm was greatly reduced. The content of O and N elements was slightly varied with the addition of Fe(NO_3_)_3_·9H_2_O; in particular, the amount of N-Q and O-C=O increased by more than two folds. The optimized sample CZnFe-5 with the addition of 5% Fe(NO_3_)_3_·9H_2_O exhibited the best capacitance performance. The specific capacitance of the CZnFe-5 electrode reached 271 F g^−1^ at 0.2 A g^−1^ and retained 133 F g^−1^ at 100 A g^−1^. The maximum energy density of CZnFe-5//CZnFe-5 symmetric SCs reached 16.83 Wh kg^−1^, and the capacitance retained 86.33% after 40,000 cycles tests at 50 A g^−1^. This work provides a novel idea for the development of high-performance and low-cost supercapacitor electrodes from biomass products.

## Figures and Tables

**Figure 1 molecules-28-06994-f001:**
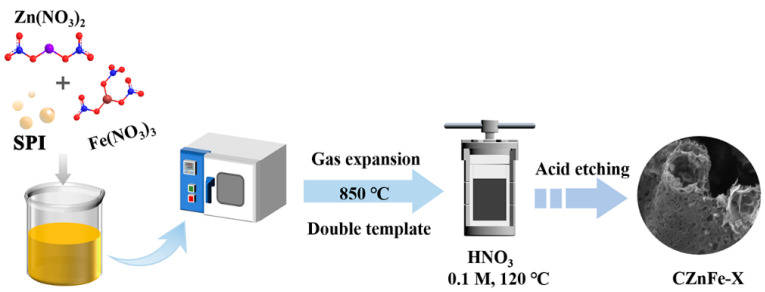
Schematic diagram of material preparation.

**Figure 2 molecules-28-06994-f002:**
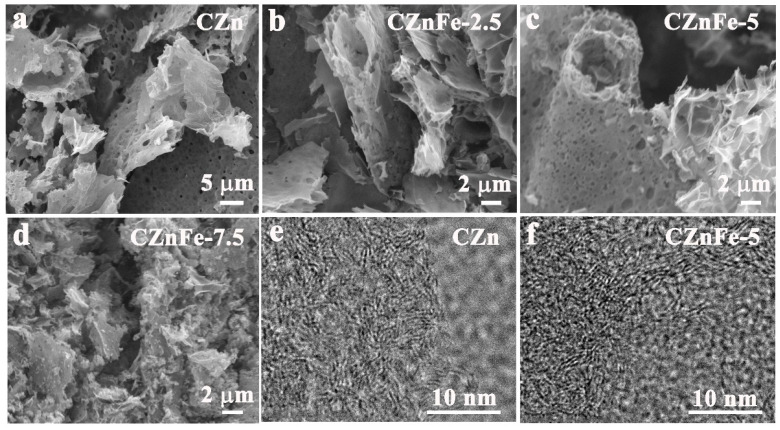
FESEM images of (**a**) CZn, (**b**) CZnFe-2.5, (**c**) CZnFe-5, and (**d**) CZnFe-7.5. The HRTEM images of (**e**) CZn, and (**f**) CZnFe-5.

**Figure 3 molecules-28-06994-f003:**
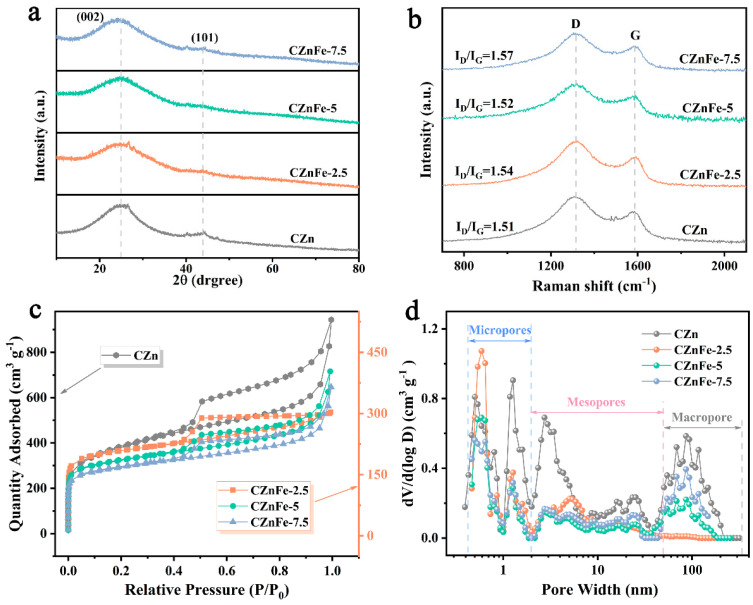
(**a**) XRD patterns and (**b**) Raman spectra of different samples, (**c**) N_2_ adsorption–desorption isotherms and (**d**) pore size distribution curves of CZn, CZnFe-2.5, CZnFe-5 and CZnFe-7.5.

**Figure 4 molecules-28-06994-f004:**
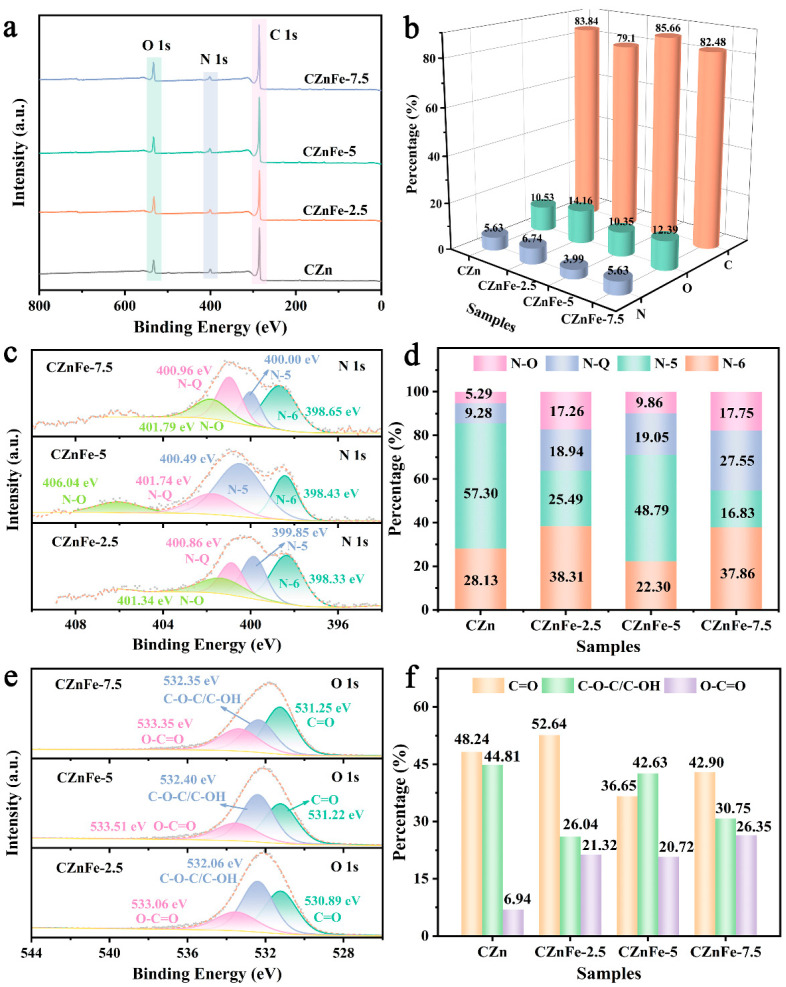
(**a**) The XPS survey spectra and (**b**) relative elemental contents of CZn, CZnFe-2.5, CZnFe-5 and CZnFe-7.5. (**c**) High-resolution spectra of N 1 s for CZnFe-2.5, CZnFe-5 and CZnFe-7.5. (**d**) Contents of N-O, N-Q, N-5 and N-6. (**e**) High-resolution spectra of O 1 s for CZnFe-2.5, CZnFe-5 and CZnFe-7.5. (**f**) The proportion of different oxygen functional groups in the samples.

**Figure 5 molecules-28-06994-f005:**
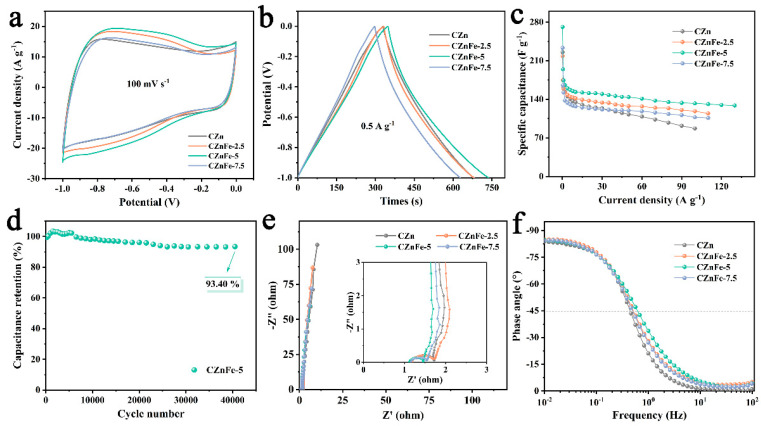
Electrochemical performance of CZn, CZnFe-2.5, CZnFe-5 and CZnFe-7.5 in a three-electrode system in 6 M KOH electrolyte: (**a**) CV curves at a scan rate of 100 mV s^−1^, (**b**) GCD curves at 0.5 A g^−1^, (**c**) the specific capacitance of different samples at various current densities, (**d**) long-cycle performances of CZnFe-5, (**e**) the comparison of EIS plots, and (**f**) Bode plots of different samples.

**Figure 6 molecules-28-06994-f006:**
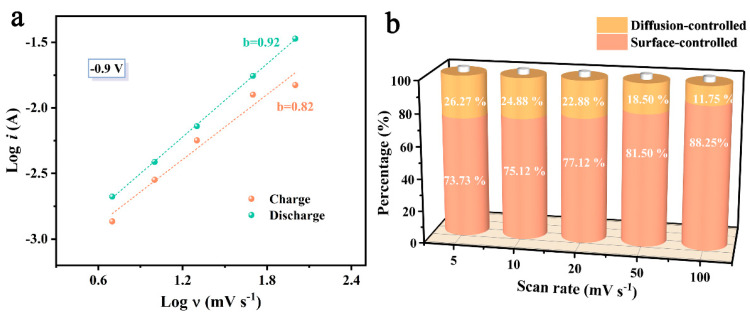
(**a**) Fitted b-value of CZnFe-5 electrode at −0.9 V from CV curves, (**b**) the percentage of the capacitive contribution at different scan rates.

**Figure 7 molecules-28-06994-f007:**
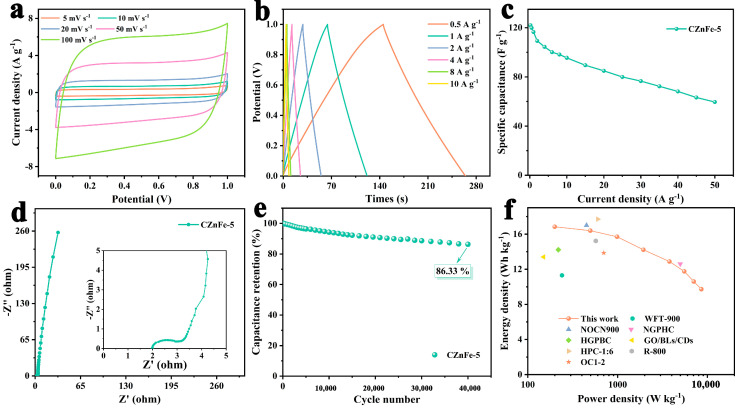
CZnFe-5 in the two-electrode system test: (**a**) CV curve at different scanning rates, (**b**) GCD curve at different current densities, (**c**) rate performance, (**d**) EIS diagram, (**e**) long-cycle performance at 50 A g^−1^ (illustration of lit device), (**f**) Ragone diagram and comparison with other reports.

**Table 1 molecules-28-06994-t001:** Summary of the pore characteristics for porous carbon products.

Sample	S_BET_ (m^2^ g^−1^)	S_micro_ (m^2^ g^−1^)	S_meso_ (m^2^ g^−1^)	D_a_ (nm)	V_t_ (cm^3^ g^−1^)	V_micro_ (cm^3^ g^−1^)	V_meso_ (cm^3^ g^−1^)	V_micro_ /V_t_	V_meso_ /V_t_
CZn	1350.34	491.66	580.53	2.99	1.01	0.21	0.64	0.21	0.63
CZnFe-2.5	775.43	538.25	171.11	2.32	0.45	0.22	0.19	0.49	0.42
CZnFe-5	616.46	377.04	169.46	2.64	0.41	0.16	0.21	0.39	0.51
CZnFe-7.5	679.70	417.74	184.41	2.69	0.46	0.17	0.23	0.37	0.50

## Data Availability

No new data were created or analyzed in this study. Data sharing is not applicable to this article.

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
