# Peer review of "High Rate Performance Supercapacitors Based on N, O Co-Doped Hierarchical Porous Carbon Foams Synthesized via Chemical Blowing and Dual Templates"

_molecules, 2023, doi:10.3390/molecules28196994_

Round 1
Reviewer 1 Report
The manuscript by Zhang et al., presents the preparation and characterization of O,N-doped carbon foams for their application in supercapacitors with aqueous electrolytes. Although there are some nice results in the work, some aspects should be clarify before it can be published:
- Have authors performed EDX to show they have removed all the metal content? Authors can also study that by XPS or EA. The assessment of the final metal content of the prepared carbons is crucial to properly characterize the samples and It be should done experimentally. The cited references are OK to explain the obtained results.
- Figure 7b shows the rate capability of the samples until 10A/g. According to this figure, the discharge time at 50A/g would be less than 2-3s. Cycling tests have been conducted to 50 A/g, which does not allow to fully see the stability of the samples and their capacity retention under more demanding conditions (lower cycling rates (i.e. at 0.5-2 A/g). Floating methods or cycling at slower rates should be conducted to better determine the capacitance retention.
-Carbon porosity data and relative doping contents doesn’t exhibit a clear trend with the increasing content of added iron. Were the samples mapped along the sample and/or with different samples? This information should be detailed on the experimental part. Error bars will give an idea of data dispersion.
-Carbon preparation seems similar to other preparations from biomass and close to the one reported by authors in ref 27. Explanation of the “blowing method” as in ref 27 would help to the reader.
Author Response
The manuscript by Zhang et al., presents the preparation and characterization of O,N-doped carbon foams for their application in supercapacitors with aqueous electrolytes. Although there are some nice results in the work, some aspects should be clarified before it can be published:
- Have authors performed EDX to show they have removed all the metal content? Authors can also study that by XPS or EA. The assessment of the final metal content of the prepared carbons is crucial to properly characterize the samples and it should be done experimentally. The cited references are OK to explain the obtained results.
Reply: Thanks for the good suggestion. We have done XPS to reveal the elements in the products. As shown in Figure 4a, there is no apparent signal of Fe in the survey spectra.
- Figure 7b shows the rate capability of the samples until 10A/g. According to this figure, the discharge time at 50A/g would be less than 2-3s. Cycling tests have been conducted to 50 A/g, which does not allow to fully see the stability of the samples and their capacity retention under more demanding conditions (lower cycling rates (i.e. at 0.5-2 A/g). Floating methods or cycling at slower rates should be conducted to better determine the capacitance retention.
Reply: Thanks for the good suggestion. Compared to other energy storage devices, the most important advantage of supercapacitor is high power density. Supercapacitor could be charge/discharge at high current density in a few seconds. The reason we choose 50 A/g is that we want to reveal the stability of as assembled supercapacitors at high current density. It is a pity that we have run out of the sample. We have kept the good suggestion in mind and will run the stability test as suggested in our future researches.
-Carbon porosity data and relative doping contents doesn’t exhibit a clear trend with the increasing content of added iron. Were the samples mapped along the sample and/or with different samples? This information should be detailed on the experimental part. Error bars will give an idea of data dispersion.
Reply: Thanks for the good suggestion. We have noticed that carbon porosity data and relative doping contents doesn’t exhibit a clear trend with the increasing content of added iron nitrate. With the addition of iron nitrate, the content of nanopores with the size larger than 1.0 nm was reduced, which leads to a sharp decrease of SSA. XPS test was performed on one spot for each sample. One batch of sample has been tested. Since the sample has been ground evenly before XPS test, the result should be reliable.
-Carbon preparation seems similar to other preparations from biomass and close to the one reported by authors in ref 27. Explanation of the “blowing method” as in ref 27 would help to the reader.
Reply: Thanks for the good suggestion. We have introduced our previous researches reported in ref. 27 (ZnCl2 as activation agent) and ref. 28 (Zn(NO3)2·6H2O as blowing agent) in the introduction part. In this research, Zn(NO3)2·6H2O was partly replaced by Fe(NO3)3·9H2O to produce highly interconnected porous carbon foam. The intermediate products originated from decomposition of Fe(NO3)3·9H2O would etch the walls of carbon foam facilitating ion transfer in the neighbor cells.
Reviewer 2 Report
After a thorough review of the submitted manuscript detailing the preparation of N, O codoped porous carbon materials for supercapacitor applications, it is with great interest that I recommend its acceptance in its present form. The authors adeptly address a recognized challenge in the field: the creation of high-performance N, O codoped porous carbon materials with a balanced pore structure. Their innovative approach, utilizing a facile chemical blowing method and employing soybean protein isolate as a self-doping carbon source, is both novel and impactful. The comprehensive investigation into the role of Fe(NO₃)₃·9Hâ‚‚O on the microstructure, elemental content, and capacitance performance of the resulting porous carbon materials is meticulously carried out, culminating in the optimized sample, CZnFe-5, which showcases impressive capacitance metrics. Furthermore, the demonstrated application in powering LED bulbs offers a tangible benefit to potential real-world usage. This work not only pushes the envelope in terms of material synthesis but also offers a sustainable alternative by capitalizing on low-cost biomass products. Given the relevance and significance of the findings, coupled with the rigorous methodologies employed, this paper undoubtedly stands as a valuable contribution to the scientific community in the realm of supercapacitor electrode materials.
Author Response
Thanks for the very positive comments!
Reviewer 3 Report
The paper deals with carbon materials from biobased materials intended for supecapacitor electrodes. The paper is well written and materials are well characterized with minor exceptions noted below. These must be addressed prior to publication:
1) Can you detect any iron in the samples?
2) The main thing which is missing in the manuscript is to provide data from a supercapacitor prepared using a reference material. The issue is that the relatively low amount of active material (2.5 mg) may lead to wrong conclusions on the material. I suggest to prepare electrodes with same loading using e.g. YP80F.
Author Response
The paper deals with carbon materials from biobased materials intended for supecapacitor electrodes. The paper is well written and materials are well characterized with minor exceptions noted below. These must be addressed prior to publication:
- Can you detect any iron in the samples?
Reply: Thanks for the good suggestion. We have done XPS to reveal the elements in the products. As shown in Figure 4a, there is no apparent signal of Fe in the survey spectra.
2) The main thing which is missing in the manuscript is to provide data from a supercapacitor prepared using a reference material. The issue is that the relatively low amount of active material (2.5 mg) may lead to wrong conclusions on the material. I suggest to prepare electrodes with same loading using e.g. YP80F.
Reply: Thanks for the good suggestion. We agree with the viewpoint. For powdery samples, the mass load of active materials is acceptable around 2 mg. For electrochemical test of each sample, we always run CV test of a few electrodes to make sure our results are reliable. The integral area of CV curves is always close to each other with the deviation about 5%. Hence, our results are reliable.
Round 2
Reviewer 1 Report
Thanks to the authors for their comments,
Yet, I would like to see the stability at lower cycling rate as it is the accurate way to determine the real suitability of the material. Stability for 40000 cycles is good, but at 50 A/g after 1 day it only retains 86% of its initial capacity (40.000cycles*2.5 s= 28h). When performing faster (the same number of cycles at 100A/g), it retains 93% of its capacity .
Authors should also show the stability at low current densities; In the case the materials still need to be improved, the reason for the capacity fading can be explained and/or a new direction for future investigations can be proposed. In this way the cualtiy of the paper and the journal will be improved.
Are the materials not reproducible or difficult to prepare?
Author Response
Thanks for the good suggestion. We have run the cycle stability test at a small current density of 2 A/g as suggested. The result reveals that the symmetric supercapacitor based on CZnFe-5 achieves 88.14% capacitance retention after 10000 cycles at 2 A g-1 (Figure S2). The cycling test confirms the good stability of CZnFe-5.

Reviewer 3 Report
The authors did not address my only main comment which I do not understand. It is rather easy to prepare electrodes from a commercial sample and it would give an idea how assembly and low loading of the supercapacitor affect its performance. I still suggest to prepare electrodes with same loading using e.g. YP80F.
Author Response
Thanks for the suggestion from the reviewer. Electrodes made from the commercial sample YP50F with close mass loading of 2.7 mg following the same procedure are used for comparison. The specific capacitance of YP50F is 71.7 F/g at the current density of 0.5 A/g and keeps 5.5 F/g at 25 A/g (Figure S1). The rate performance of CZnFe-5 is better than that of the commercial sample YP50F with close mass loading.
